# Identification of Molecular Subtypes of Clear-Cell Renal Cell Carcinoma in Patient-Derived Xenografts Using Multi-Omics

**DOI:** 10.3390/cancers17081361

**Published:** 2025-04-18

**Authors:** Zhengyuan Qiu, Dalin Zhang, Fernando Jose Garcia-Marques, Abel Bermudez, Hongjuan Zhao, Donna M. Peehl, Sharon J. Pitteri, James D. Brooks

**Affiliations:** 1Department of Urology, Stanford University School of Medicine, Stanford, CA 94305, USA; peterqiu@stanford.edu (Z.Q.); dzh@stanford.edu (D.Z.); hongjuan@stanford.edu (H.Z.); 2Department of Radiology, Stanford University School of Medicine, Stanford, CA 94305, USA; fjgarcia@stanford.edu (F.J.G.-M.); abermu50@stanford.edu (A.B.); spitteri@stanford.edu (S.J.P.); 3Canary Center at Stanford for Cancer Early Detection, Stanford University School of Medicine, Stanford, CA 94305, USA; 4Department of Radiology and Biomedical Imaging, University of California at San Francisco, San Francisco, CA 94305, USA; donna.peehl@ucsf.edu; 5Center of Academic Medicine, 453 Quarry Road, Palo Alto, CA 94304, USA

**Keywords:** patient-derived xenografts, renal cell carcinoma, molecular subtypes

## Abstract

Clear-cell renal cell carcinoma (ccRCC), the most common form of renal cell carcinoma, is a heterogenous disease with multiple molecular subtypes identified through profiling the genetic makeup of each tumor. This heterogeneity among ccRCC patients presents considerable challenges for developing universally effective treatment strategies. Patient-derived xenograft (PDX) models are the most realistic pre-clinical platform in understanding the biology of ccRCC, identifying novel targets for drug discovery, and testing new therapeutic agents. However, PDXs representing distinct molecular subtypes of ccRCC have not been classified. We previously established and characterized ccRCC PDXs using a multi-omics approach. Here, we demonstrated that each of our PDX models closely resembles one of the molecular subtypes in terms of gene expression and metabolism, suggesting that proper PDX subtypes should be used when investigating the corresponding ccRCC patient subtypes. This “matching” strategy will greatly facilitate the clinical translation of positive findings into the optimal management of ccRCC patients.

## 1. Introduction

The incidence of renal cell carcinoma (RCC) has been increasing by about 3% per year in the United States, with an estimated 80,980 newly diagnosed cases and 14,510 deaths in 2025 [1]. Among the various histological subtypes of RCC, clear-cell renal cell carcinoma (ccRCC) is the most prevalent, representing around 70–80% of cases [2]. ccRCC is characterized by the presence of clear cytoplasm in tumor cells, resulting from the accumulation of lipids and glycogen [3]. The pathogenesis of ccRCC is closely associated with genetic mutations, particularly in the von Hippel–Lindau (VHL) gene, which is inactivated in more than 90% of cases [3]. VHL inactivation leads to the stabilization of hypoxia-inducible factors (HIFs), promoting the transcription of genes involved in angiogenesis, such as vascular endothelial growth factor (VEGF), and driving tumor progression [4]. Similar to most solid tumors that are heterogenous at initiation and progression [5], ccRCC tumors show different molecular profiles, treatment responses, and clinical outcomes, even though they have common histopathology and hallmark genetic mutations [3,6].

Previous studies have identified molecular subtypes of ccRCC with differential treatment responses and clinical outcomes in multiple patient cohorts by multi-omics profiling. For example, the TRACERx study defined seven major clonal subtypes of ccRCC based on genetic alterations: VHL mono-driver, PBRM1-SETD2, PBRM1-somatic copy number alteration (SCNA), PBRM1-PI3K, VHL wildtype, multiple clonal driver, and BAP1 driven [7]. Transcriptomic profiling by bulk RNA sequencing of >1500 ccRCC tumor samples from the IM151 and JR101 clinical trials identified seven molecular subtypes with differential outcomes after immunotherapy and targeted therapy using a machine learning model [8], while single-cell RNA sequencing uncovered eleven ccRCC subpopulations, including three cytotoxicity-related molecular subtypes [9]. Proteomic profiling revealed ccRCC subtypes with increased expression of proteins related to glycolysis, mitochondria, translation, nucleosome, and spliceosome, or a higher expression of proteins involved in focal adhesion, extracellular matrix, and collagen organization [10]. Multi-omics analysis integrating gene expression alteration, as well as genetic and epigenetic changes, identified three major subtypes with immune cold, hot, or exhausted tumor microenvironment that may respond differently to immunotherapy [11]. Another proteogenomic analysis classifies a Chinese cohort of ccRCC into three subtypes, among which the most aggressive subtype exhibits the strongest immune phenotype, increased metastasis, and metabolic imbalance [12]. Combined histopathologic, proteogenomic, and metabolomic analyses revealed an aggressive histopathologic subtype with high tumor grade, BAP1 mutation, genome instability, increased hypermethylation, and a specific protein glycosylation signature [13]. Subtypes with differential prognosis based on molecular signatures related to G2M checkpoint, glycosylation, senescence, hypoxia, pyroptosis, differentiation, angiogenesis, metabolism, collagen, stemness, and neutrophil extracellular traps have also been reported [14,15,16,17,18,19,20,21,22,23,24]. The complexity of the heterogeneous nature of ccRCC creates significant challenges to the understanding of molecular mechanisms underlying ccRCC progression and the development of effective therapies for the optimal management of ccRCC patients.

We and others have demonstrated that patient-derived xenografts (PDXs) are the optimal models of human ccRCC for the investigation of tumor progression mechanism, therapy response, and drug resistance [25,26,27]. In particular, we applied tissue slice methodology to improve the survival and growth of ccRCC PDXs by implanting precision-cut 300 μm thick tissue slices under the renal capsule of RAG2^−/−^γC^−/−^ mice, widely used for stem cell studies because they are highly tolerant of human cells and the strain we use to establish PDXs from ccRCC tissues [28]. We demonstrated that these PDXs maintained similar genotypes and immunohistologic phenotypes to the parental tumors [29]. In addition, these ccPDXs have shown the capacity to produce metastatic disease from primary orthotopic or subcutaneous tumors with gross metastases to the lung, liver, and bone [29]. They also responded to standard-of-care tyrosine kinase inhibitors (TKIs) and mTOR inhibitors (sunitinib, cabozantinib, and temsirolimus) as expected [29,30,31], demonstrating that these PDX models are authentic models, ideal for developing novel therapies for ccRCC. This strategy of using multiple PDXs with different characteristics, which reflects a traditional phase II clinical trial without pre-selection for a particular tumor characteristic, will greatly improve our ability to generalize the results obtained due to enhanced heterogeneity and facilitate their translation to clinical settings.

Although the consensus is that PDXs retain the individual characteristics of parental ccRCC tissues better compared to traditional commercial and mature RCC cell lines, whether the molecular subtypes identified in ccRCC patient samples are preserved in PDX models is not clear. The identification of molecular subtypes that represent human ccRCC subtypes is vital for the investigation of the different mechanisms of tumor progression, treatment response, and drug resistance with respect to the different unique molecular subtypes. This is important because proper PDX subtypes should be used when investigating the corresponding ccRCC patient subtypes. This “matching” strategy will greatly facilitate the clinical translation of positive findings into the optimal management of ccRCC patients. Here, we compared the transcriptomic and proteomic profiles of five ccRCC PDX models established in our lab to those of the human ccRCC molecular subtypes reported by our group, as well as other groups. In addition to ccRCC PDXs, xenografts derived from cells cultured from four of the five PDXs (XENs) were also included, since they retain the morphologic and metabolic features of the corresponding parental PDXs and represent realistic pre-clinical models for ccRCC [32]. Hierarchical analysis, Principal Component Analysis (PCA), and Permutation Correlation Analysis were performed to identify molecular subtypes in ccRCC patients that the PDXs and XENs represent. Enrichment of key molecular pathways in subtypes of ccRCC PDXs and XENs was determined using Gene Set Enrichment Analysis (GSEA).

## 2. Materials and Methods

### 2.1. Characteristics of ccRCC PDXs

In this study, we used multi-omics data for five PDXs established in our lab from high-grade primary or metastatic ccRCC specimens obtained from patients undergoing nephrectomy or autopsy between September 2011 and May 2012 at Stanford University under an institutional review board-approved protocol with informed consent as previously described [28]. We selected five PDXs that had multi-omics data including transcriptomic, proteomic, and metabolic profiles for this study. The clinicopathological characteristics of the parental tumors are listed in Table 1.

### 2.2. Publicly Available Datasets Used in This Study

Bulk RNA sequencing data for the five PDXs were downloaded from Gene Expression Omnibus (GEO Accession: GSE213857) [32]. Transcript counts were normalized to the total number of transcripts detected in each sample. The mouse sequences were computationally removed from the PDX for the transcriptomic analysis using the R-package XenofilteR [33] available at https://github.com/NKI-GCF/XenofilteR (accessed on 6 March 2024). Only genes with an average expression taken across all samples greater than one transcript per million were selected for further analyses. Significantly differentially expressed genes were identified using the R packages DEGseq2 (version 1.30.1) (Appendix A).

Similarly, gene expression profiling data from our previously published dataset of 177 ccRCCs from European patients was used to build a model of 2 major subtypes that differed in outcomes (GEO accession number GSE3538) [34]. We selected 3674 genes represented by 5560 clones on the microarrays whose expression was both well measured and highly variable among samples, as previously described for further analyses (Appendix A) [34]. Briefly, well-measured genes were defined as those with a ratio of signal intensity to background noise of more than 1.5 for either the Cy5-labeled cRCC sample or the Cy3-labeled reference sample in at least 70% of the samples hybridized; genes with highly variable expression were defined as those whose expression was higher or lower by a factor of at least three than the average expression of all cRCC samples in at least ten cRCC samples.

For the 7 molecular subtypes represented in the IM151 and JR101 trials, the levels of average expression of each of the 19,401 genes across all cases within each subtype were downloaded from https://ars.els-cdn.com/content/image/1-s2.0-S1535610824000874-mmc2.xlsx (accessed on 27 August 2024) and used for further analyses.

Transcriptomic and proteomic profiles of 110 ccRCC tissues in the CPTAC ccRCC Discovery and Confirmatory Study (PDC000200 version 2) were downloaded from Proteomic Data Commons (https://proteomic.datacommons.cancer.gov/pdc/browse/filters/primary_site:Kidney) (accessed on 13 September 2024).

### 2.3. Proteomic Profiling of ccRCC PDX Tissues by Liquid Chromatography–Mass Spectrometry (LC-MS)

Protein extraction from tumor tissues was performed as previously described [35]. Protein concentrations were determined by BCA protein assay according to the manufacturer’s instruction (Thermo Fisher Scientific, Waltham, MA, USA). A total of 25 µg of protein was reduced, alkylated, and digested with trypsin (Thermo Fisher Scientific) as previously described [35]. The resulting tryptic peptides were reconstituted in 0.1% formic acid (50 µL, Fisher Scientific). The tryptic peptides were fractionated using a Dionex Ultimate Rapid Separation liquid chromatography system (Thermo Fisher Scientific) and analyzed on an Orbitrap Tribrid Eclipse mass spectrometer, equipped with an Orbitrap mass analyzer (Thermo Fisher Scientific, Waltham, MA, USA), as previously described [35].

### 2.4. ccRCC PDX Proteomic Data Processing

The collected raw data were processed through a two-stage search using Byonic 4.0.12. The initial stage involved a search against the Swiss-Prot database consisting of the reference human proteome (as of 2022; 20,645 entries). This was followed by a second-stage search, which utilized the Swiss-Prot database containing the mouse reference proteome (as of 2022; 17,380 entries). The search parameters were set to include trypsin digestion, allowing for a maximum of two missed cleavages, a precursor mass tolerance of 0.5 Da, and a fragment mass tolerance of 10 ppm. The parameters also specified fixed cysteine carbamidomethylation, along with variable modifications such as methionine oxidation and asparagine deamination. To maintain the integrity of the peptide identification process, peptides with a false discovery rate (FDR) greater than 1% were excluded from the results. Furthermore, to focus on non-homologous peptides, any peptides identified in both the human and mouse database searches were discarded. A meticulous analysis of the human-identified proteins versus homologous mouse peptides was performed using a custom R script. This included three biological replicates for each tissue xenograft from three RCC patients and one control. Protein concentrations were determined by evaluating the unique signals associated with each protein within the samples. The relative protein abundance was ascertained by comparing these specific signals to the mean signal observed across all samples, thus providing an estimate of each protein’s relative abundance within the complete protein content of the experimental dataset. A normalization procedure was carried out to ensure comparability across samples and to enable statistical analysis. This process adjusted the relative abundance values to a normal distribution with a mean of 0 and a standard deviation of 1. Raw data for the five ccRCC PDX tissues have been deposited to the ProteomeXchange Consortium via the PRIDE partner repository (https://www.ebi.ac.uk/pride/) (accessed on 21 February 2025) with the dataset identifier PXD061075. Protein expression levels used for hierarchical cluster analysis were provided in Appendix A.

### 2.5. ccRCC PDX Metabolic Data

Absolute concentrations of the metabolites (nmol/mg) in the five PDXs were obtained from a previous study [32], in which the metabolic labeling of PDXs, extraction of metabolites, and measurement of the concentration of metabolites were described in detail. Metabolite levels in the PDX subtypes used for comparison are provided in Appendix A.

### 2.6. Comparison Methods

Hierarchical analysis, PCA, and Pearson Correlation Analysis, as well as graph generation, were carried out using Python version 3.12.4. Missing values were imputed using a KNNImputer algorithm (https://scikit-learn.org/stable/modules/impute.html) (accessed on 18 March 2024).

### 2.7. GSEA

The R package “org.Hs.eg.db” was used for ID transform and “clusterProfiler” was used for KEGG pathway analysis with differentially expressed genes between cluster 1 and 2 (R software 4.4.1). GSEA was conducted using “c2.cp.kegg.v2022.1.Hs.symbols.” in GSEA software (version 4.3.3).

### 2.8. Statistical Analysis

Statistical analyses were performed using GraphPad Prism 10.2.3. All error bars represent the mean ± SEM. Notably, *p*-values are indicated as * *p* < 0.05; ** *p* < 0.01; *** *p* < 0.001; **** *p* < 0.0001; NS, *p* > 0.05 is not significant.

## 3. Results

### 3.1. Study Design

The transcriptomics and proteomics datasets for ccRCC patient and PDX samples were filtered to exclude genes with more than 50% missing data across each dataset. We first performed unsupervised hierarchical analysis using ccRCC PDX and XEN transcriptomic profiles to determine whether XENs retain parental PDX transcriptional programs. Principal Component Analysis (PCA) and Pearson Correlation Analysis were then employed to determine which molecular subtype of the 177 ccRCC patients is best represented by each PDX. We also performed Pearson Correlation Analysis to determine which of the seven molecular subtypes represented in the IM151 and JR101 trials each PDX most closely resembles. Moreover, we identified two subtypes in CPTAC patients based on transcriptomic profiles using hierarchical clustering and determined the correlation of each ccRCC PDX with CPTAC subtypes at both transcript and protein levels using PCA. Finally, we compared the levels of metabolites involved in glycolysis and KEGG pathways enriched in the two main ccRCC PDX subtypes. This workflow is depicted in Figure 1.

### 3.2. Transcriptomic Programs Are Retained in ccRCC PDXs with Different Passages and Between PDX and Their Respective XEN Counterparts

Hierarchical clustering analysis of bulk RNA sequencing data revealed that each PDX tumor and corresponding XEN grouped together in the same cluster, regardless of their passages (Figure 2A). Notably, samples 054 and 047 were positioned further from the other samples in the cluster, indicating greater divergence. This suggests that these two samples show a larger difference compared to the others. Overall, the hierarchical analysis of the five xenografts demonstrated a strong consistency between the PDX and cell-cultured PDX results, with all samples falling into the same cluster. This clustering indicates that the transcriptomic profiles are strongly associated with the patient-specific characteristics, reinforcing the relationship between the data and their clinical origins. Among these, sample 054 displayed the most distinct difference, while samples 068 and 093 were the most similar to each other.

### 3.3. Transcriptomic Programs in ccRCC PDXs Resemble Molecular Subtypes of ccRCC Patients

We compared the transcriptional profiles of each PDX tumor and the corresponding XEN with the two major molecular subtypes of ccRCC that we previously identified in a cohort of 177 Swedish patients treated at Umea Hospital (GSE 3538) (Figure 2B) [34]. Consistent with hierarchical clustering analysis, PCA demonstrated that PDXs and XENs from 093, 068, and 072 were positioned closer together, indicating higher similarity in transcriptional profiles among them, while PDXs and XENs from 047 and 054 were more distant from them, suggesting distinct gene expression programs present in these two lines compared to 093/068/072 (Figure 2C). Moreover, 093/068/072 overlapped significantly with type I of the GSE3538 cohort, while 054 and 047 were mostly overlapped with type II in the PCA plot, indicating similarity between these two lines with type II of the GSE3538 cohort (Figure 2C). Notably, 072 and 047 were positioned apart at the border of GSE3538 type II samples, suggesting they possess differential gene expression programs even though they represent the same major subtype of the GSE3538 cohort.

We next determined whether the two molecular subtypes observed in GSE3538 cohort are recapitulated in an independent dataset, i.e., a cohort of 110 ccRCC patients in the CPTAC ccRCC Discovery and Confirmatory Study (PDC000200 version 2). The transcriptomic profiles of these 110 ccRCC tumors downloaded from Proteomic Data Commons were grouped into two main subtypes, i.e., cluster 1 and 2, by hierarchical clustering analysis (Figure 3A). PCA demonstrated the cluster 1 primarily overlapped with type I in GSE3538, while cluster 2 mainly overlapped with type II in GSE3538 (Figure 3B), suggesting that these two molecular subtypes were common in different cohorts of ccRCC patients. More importantly, PCA showed that 093/068/072 overlapped significantly with cluster 1 of the CPTAC cohort, while 054 and 047 mostly overlapped with cluster 2, indicating similarity between subtypes of PDXs with those of ccRCC patients (Figure 3C). Consistently, Pearson correlation analysis demonstrated gene expression levels of 093/068/072 correlated with cluster 1 to a higher degree, while 054 and 047 with cluster 2 (Figure 3D). These results suggest that transcriptomic programs in distinct subsets of PDX models resemble molecular subtypes of ccRCC patients.

Finally, we examined the correlation of the expression profiles of our PDXs with those of the molecular subtypes represented in the IM151 and JR101 trials [8]. Interestingly, both 068 and 092 correlated with subtype 2 > subtype 5 > subtype 1, 047 correlated with subtype 4 > subtype 6 ≈ subtype 5, and 054 correlated with subtype 3 > subtype 6 ≈ subtype 5, while 072 correlated with subtype 5 ≈ subtype 2 (Figure 4A). Subtype 7 was not included in the comparison due to the small number of differentially expressed genes found in our PDX dataset. On the other hand, both subtypes 1 and 2 were best resembled by 093 > 068, subtype 3 by 054, subtype 4 by 047 > 054, subtype 5 by 093 ≈ 072, and subtype 6 by 047 ≈ 054. These results indicate that each PDX line represents a particular subtype of ccRCC patients better than the other subtypes, and each subtype of ccRCC patients resembles a particular PDX line more closely than the other lines, which has important implications when choosing PDX lines to investigate the molecular subtypes of ccRCC.

### 3.4. Different Pathways Are Enriched in PDX Subtypes

We performed GSEA to identify the pathways enriched in differentially expressed genes between 093/068/072 vs. 054/047. The enrichment factor, false discovery rate (FDR), and gene counts for each enriched pathway identified are listed in Table 2. Multiple pathways involved in adhesion and extracellular matrix (ECM) modeling, including cell adhesion, focal adhesion, and ECM–receptor interaction (Figure 4B), were upregulated in 093/068/072, suggesting subtype 1 represents tumors that are highly active in tissue reorganization. In contrast, 054/047 showed high activities of immune response-related pathways, including complement and coagulation cascades, cytokine–cytokine receptor interaction, TNF signaling, viral protein interaction with cytokine, and IL-17 signaling (Figure 4C), suggesting that subtype 2 encompasses immune-hot tumors. We added this to the result section. The fold enrichment of these pathways ranges from 2 to 5 (Figure 4D). ECM–receptor interaction and complement and coagulation cascades are the top pathways with the highest fold enrichment and the lowest FDRs, confirming the significance of these pathways in distinguishing the two PDX subtypes.

### 3.5. Proteomic Programs in ccRCC PDXs Resemble Molecular Subtypes of ccRCC Patients

We performed LC-MS-based proteomic analysis on 45 PDX tissues from the five lines. Hierarchical clustering analysis was performed using 1376 proteins well measured in at least 50% of the samples (Appendix A). Similarly to what we observed in hierarchical clustering of transcriptional profiles of these PDX lines, 093 and 068 displayed a high degree of similarity and clustered together (Figure 5A). In addition, the 054 protein expression profile differed substantially from 093/068 and clustered at the opposite side of the dendrogram (Figure 5A). Unlike transcriptional profiles, 047 showed a protein expression pattern closer to 093/068 compared to 054, while the majority of 072 samples clustered with 054 (Figure 5A). In other words, three PDX lines display significant differences in both protein and transcript expression that distinguishes the two major subtypes in human ccRCC. Not surprisingly, the two PDX lines with relatively weak correlation to the human subtypes can change their cluster assignment based on their transcriptomic and proteomic profiles.

We next compared the proteomic profiles of PDX lines to those of the 110 ccRCC patients in the CPTAC cohort. PCA demonstrated that 093 and 068 largely overlapped with cluster 1 of CPTAC, while 054, 072, and 047 overlapped with cluster 2, consistent with patterns observed in transcriptomic profiles (Figure 5B). In addition, Pearson correlation analysis showed higher correlation between 093/068/072 and cluster 1, as well as between 054/047 and cluster 2 (Figure 5C), confirming the differential representation of ccRCC patient subtypes by PDX subtype at the protein level.

### 3.6. PDX Subtypes Showed Significant Differences in Metabolism

We compared the levels of key metabolites between 093/068/072 as type 1 and 054/047 as type 2. The absolute concentrations of metabolite levels revealed significant differences between type I and type II PDX models (Figure 6). Both large and small metabolism indicators, such as lactate and leucine, were significantly higher in type 2 PDXs compared to type 1. These pronounced differences indicate distinct metabolic programs between the two PDX subtypes.

## 4. Discussion

We have compared the transcriptional profiles of our PDX models to those of ccRCC patients and demonstrated that transcriptomic programs in ccRCC PDXs resemble the molecular subtypes of ccRCC patients in multiple independent cohorts. In addition, proteomic profiling revealed that ccRCC patient subtypes are recapitulated by PDX subtypes at the protein level. These PDX subtypes not only differed in transcriptomic and proteomic profiles but also displayed significant differences in key metabolism indicators. Our study revealed both similarities and distinctions between the molecular profiles of PDX subtypes and corresponding ccRCC patient subtypes, suggesting that proper PDX subtypes should be used when investigating the corresponding ccRCC patient subtypes. This “matching” strategy will greatly facilitate the clinical translation of positive findings into the optimal management of ccRCC patients.

PDX models have served as a vital in vivo model and have been extensively employed in cancer research because they maintain the genomic and pathological characteristics of patients, replicate tumor heterogeneity, and faithfully mirror clinical responses to drugs [36]. Moreover, multi-omics studies have shown that molecular subtypes in patients are represented by PDX models in several cancer types. For instance, genomic and proteomic landscapes of subtypes of small-cell lung cancer (SCLC) tissues are preserved in the derivative PDX models, offering a powerful system to characterize SCLC biology and inform clinical research treatment strategies for patients with different subtypes of SCLC [37]. Similarly, gene expression at both the transcriptomic and proteomic levels of breast cancer PDX models are comparable to those contained in the TCGA and CPTAC repositories, where they represent all of the clinical subtypes, as well as many of the molecular subtypes [36]. Our study indicates that PDX models of ccRCC also retain the characteristics of the molecular subtypes identified in patients at the transcriptomic and proteomic levels, underscoring the translation relevance of these models in the identification of new treatments, mechanisms of drug resistance, and predictive biomarkers in ccRCC.

Genetic mutations play a key role in the development, progression, prognosis, and treatment response of clear-cell renal cell carcinoma (ccRCC) [38]. A recent study of 252 ccRCC patients by Ruan et al. identified two molecular subtypes based on gene expression, DNA methylation, and gene mutation data [39]. Although no significant difference was observed for the mutation frequency of the most frequently mutated gene VHL between the two subtypes, other frequently mutated genes, including PBRM1 and BAP1, were differentially mutated. In addition, no significant difference in TMB was observed between the two subtypes. We are in the process of generating genetic mutational profiles for our PDX models. It would be interesting to determine the mutational similarity of our PDX subtypes to ccRCC patient subtypes, such as the negative correlation between the presence of PBRM1 and BAP1 mutants.

Lactate metabolism plays a critical role in ccRCC progression, where tumor cells preferentially rely on glycolysis rather than oxidative phosphorylation for energy production, resulting in lactate accumulation even in the presence of oxygen [40]. Lactate also serves as a signaling molecule that activates oncogenic pathways and promotes angiogenesis by stabilizing HIF, further driving tumor progression [41]. Lactate can inhibit the function of immune cells, such as T cells and natural killer cells, promoting immune evasion and creating an immunosuppressive environment that supports tumor growth [42]. A high lactate level in subtype 2 tumors indicates the potential to target lactate metabolism as a therapeutic strategy to disrupt the metabolic and signaling pathways that sustain their growth and progression, as well as potentiate anti-angiogenic and immunotherapy. Interestingly, subtype 2 also showed a high level of branched-chain amino acids (BCAAs), including leucine, isoleucine, and valine, which have been shown to have double-edge effects on cancer progression [43,44]. On the one hand, BCAAs are indispensable for maintaining immune system function, and dietary supplementation with BCAAs can enhance the activity of immune cells [45,46]. For example, leucine is essential for the upregulation of mTORC1 during T cell activation [47]. On the other hand, BCAAs can serve as an energy source and an activator of oncogenic signaling pathways, potentially fueling cancer development by promoting cell proliferation, migration, and invasion [48]. Therefore, the utility of BCAAs, including Leu, in ccRCC, either as biomarkers or as potential therapeutic targets, will depend on further research clarifying their effects.

A common concern is that PDX models may under-represent tumor heterogeneity due to sampling bias resulting from spatial heterogeneity, differential capacities for engraftment, and tumor evolution and selection during PDX passages. For instance, Hoge et al. demonstrated that later-passage PDXs exhibit significantly reduced genomic similarity, e.g., DNA copy number, to their parental tumors compared with earlier-passage PDXs [49], indicating ongoing divergence of genomic landscapes during model propagation. Our results revealed differences in transcript and protein expression profiles among different passages of the same PDX line, albeit the magnitude of the changes was limited. In particular, proteomic analysis showed that one-third of 072 samples clustered together and displayed slightly different patterns from the other samples that clustered next to them. These observations together with the lack of human immune system and the replacement of stromal cells by host cells add a note of caution in interpretating findings generated from PDXs in cancer research. However, these limitations can be overcome in part by using humanized mice and early-passage PDXs. Moreover, integrating omics technologies with PDX models enables ongoing monitoring of tumor heterogeneity and specific characteristics like clonal metastasis potential.

In several malignancies, differences in the biology of different molecular subtypes can be used for prognostication and therapy selection, as in the Her2 subtype of breast cancer. As targeted therapies in ccRCC expand and improve, it is likely that therapeutic approaches could be tailored to each subtype. Our finding that PDXs correlate with distinct molecular subtypes will help with selection of the optimal in vivo models for each tumor subtype. The matching strategy will allow for more accurate pre-clinical testing and personalized treatment strategies in ccRCC patient care. Different PDX subtypes are ideal models for identifying effective treatments and biomarkers for treatment responses for each patient subtype they resemble. In the real world, when a patient is assigned to a molecular subtype, they can be treated with the most effective treatment(s) identified in the same PDX subtype. In addition, PDX subtypes can be used to test novel drugs and delivery methods to identify effective new treatments for corresponding ccRCC patient subtypes. When multiple PDX lines in the same subtype are used, this mimics a clinical trial conducted in the corresponding ccRCC patient subtype. These applications will greatly facilitate the clinical translation of findings and improve ccaRCC patient outcomes. For example, an efficacy study with an HIF2 or VEGF inhibitor would be appropriate to determine whether the two PDX subtypes respond differently and whether PDX lines within the same subtype would respond similarly. It would also be interesting to test whether the two subtypes respond differently to other standard-of-care treatments such as immune checkpoint inhibitors. Our results provide a strong rationale for such studies in the future, which may help the research community to select the right models for investigating mechanisms of treatment response and resistance in ccRCC. Ultimately, PDXs hold promise for accelerating the discovery of new drugs, enhancing our understanding of disease progression and resistance to therapies, and advancing personalized treatment approaches.

## 5. Conclusions

PDX models are the most realistic pre-clinical platform in understanding the biology of ccRCC, identifying novel targets for drug discovery, and testing new therapeutic agents. We demonstrated that each of our PDX models resembles one of the molecular subtypes in gene expression at the transcript and protein levels, suggesting that proper PDX subtypes should be used when investigating the corresponding ccRCC patient subtypes. Our study will greatly facilitate the clinical translation of positive findings to the optimal management of ccRCC patients.

## Figures and Tables

**Figure 1 cancers-17-01361-f001:**
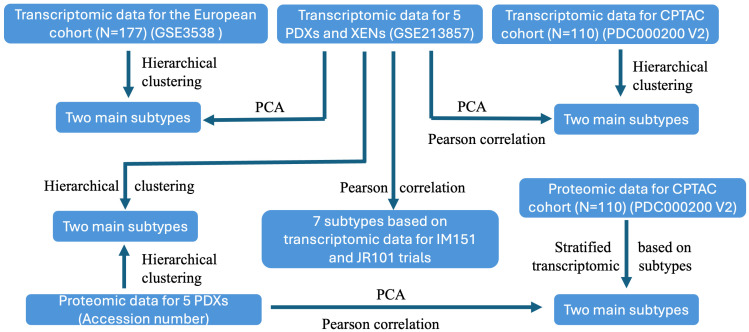
The schematics of our study design.

**Figure 2 cancers-17-01361-f002:**
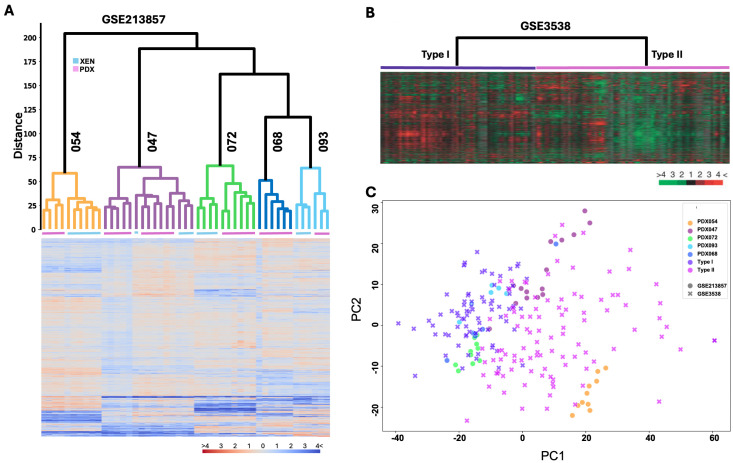
Comparison of transcriptomic profiles of PDX models to ccRCC patients in GSE3538. (**A**) Hierarchical clustering analysis of transcriptomic profiles of tumor tissues from five PDX lines and xenografts derived from cells cultured from four of the five PDXs (XENs); Orange color indicates high expression; blue indicates low expression; (**B**) hierarchical clustering analysis of transcriptomic profiles of 177 ccRCC patients in GSE3538; (**C**) PCA of samples from 5 PDX lines and 177 ccRCC patients.

**Figure 3 cancers-17-01361-f003:**
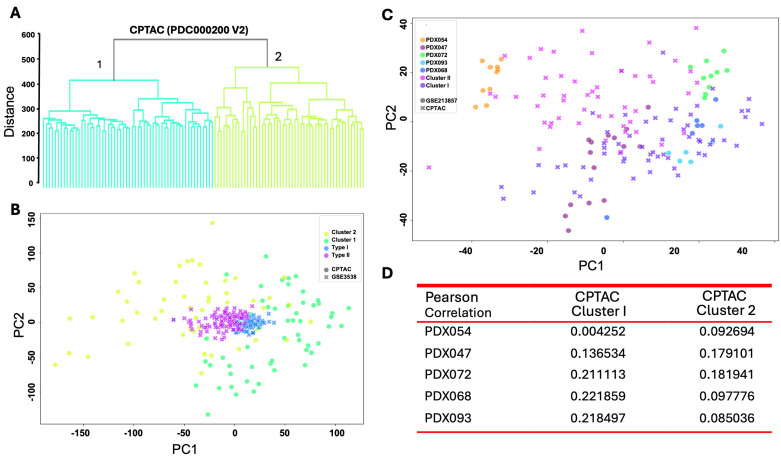
Comparison of transcriptomic profiles of PDX models with ccRCC patients in the CPTAC ccRCC Discovery and Confirmatory Study. (**A**) Hierarchical clustering analysis of transcriptomic profiles of tumor tissues from 110 ccRCC patients; (**B**) PCA of CPTAC and GSE3538 datasets; (**C**) PCA of transcriptomic profiles of 5 PDX lines and CPTAC dataset; (**D**) Pearson correlation analysis of transcriptomic profiles of 5 PDX lines and CPTAC dataset.

**Figure 4 cancers-17-01361-f004:**
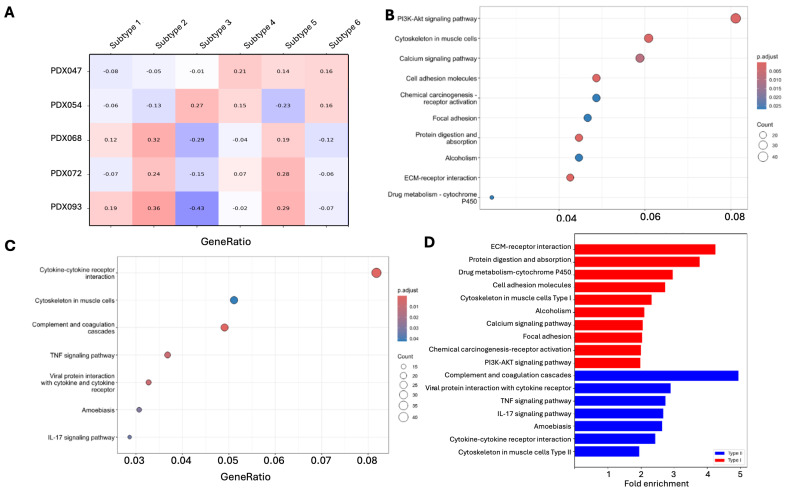
GSEA to identify pathways enriched in differentially expressed genes between PDX subtypes. (**A**) Pearson correlation analysis of expression profiles of 5 PDXs with those of the molecular subtypes represented in IM151 and JR101 trials; (**B**) pathways upregulated in 093/068/072; (**C**) pathways upregulated in 054/047; (**D**) fold enrichment of each pathway identified in (**B**,**C**).

**Figure 5 cancers-17-01361-f005:**
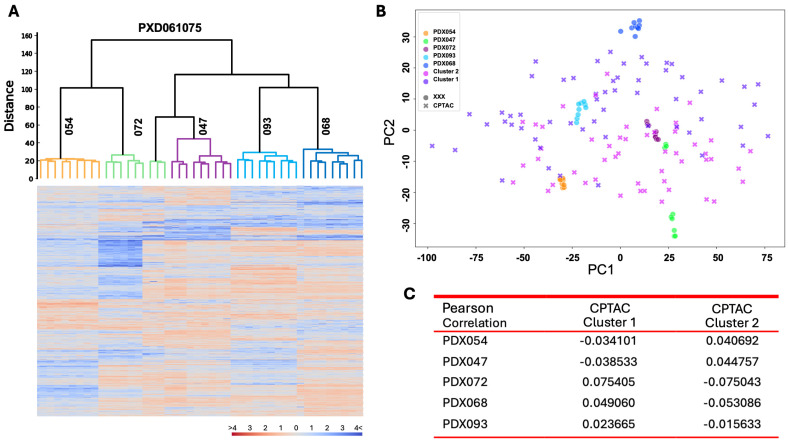
Comparison of proteomic profiles of PDX models to ccRCC patients in the CPTAC ccRCC Discovery and Confirmatory Study. (**A**) Hierarchical clustering analysis of proteomic profiles of tumor tissues from 5 PDX lines; Orange color indicates high expression; blue indicates low expression; (**B**) PCA of proteomic profiles from 5 PDXs and CPTAC datasets; (**C**) Pearson correlation analysis of proteomic profiles of 5 PDX lines and CPTAC datasets.

**Figure 6 cancers-17-01361-f006:**
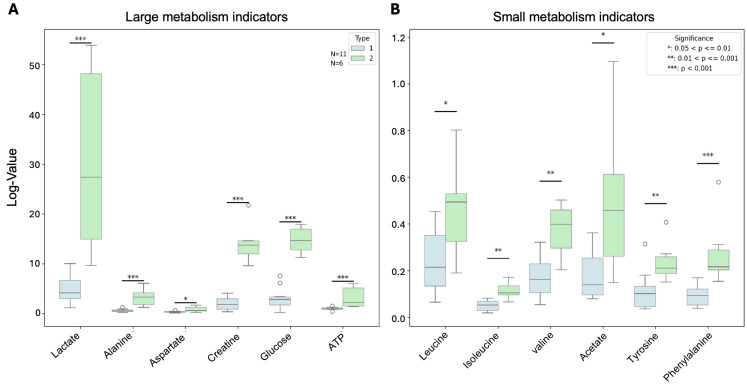
Comparison of the levels of key metabolites between PDX subtypes by Student’s *t*-test. Both large (**A**) and small (**B**) metabolism indicators were significantly higher in type 2 PDX (N = 6) compared to type 1 (N = 11).

**Table 1 cancers-17-01361-t001:** Clinicopathological characteristics of parental tumors from which the PDXs were derived.

PDX ID	Age	Gender	Furhman Grade	Clinical Stage	Source of Tissue	Previous Treatment	VHL Status
PDX047	72	Female	IV	TXNXM1	Primary tumor	Radiation and chemotherapy	Mutated
PDX054	44	Male	IV	pT3aN0M1	Colon metastasis	Chemotherapy	Mutated
PDX068	74	Male	III	pT3aN0M1	Primary tumor	None	Mutated
PDX072	65	Female	IV	TXNXM1	Lung metastasis	Immunotherapy, radiation, and chemotherapy	Wild type
PDX093	73	Female	IV	pT3bNXM1	Adrenal gland metastasis	None	Mutated

**Table 2 cancers-17-01361-t002:** Fold Enrichment of each pathway upregulated in type I or type II ccRCC PDXs.

**Pathway Upregulated in Type I**	**Rich Factor**	**Fold Enrichment**	**FDR**	**Count**
PI3K-AKT signaling pathway	0.110497238	1.987157218	0.001347240	40
Cytoskeleton in muscle cells Type	0.129310345	2.325487865	0.000952201	30
Calcium signaling pathway	0.114173228	2.053265401	0.008726537	29
Cell adhesion molecules	0.151898734	2.731712327	0.000624878	24
Chemical carcinogenesis-receptor activation	0.111627907	2.007490919	0.026983909	24
Focal adhesion	0.113300493	2.037570319	0.026983909	23
Protein digestion and absorption	0.20952381	3.768028591	7.75207 × 10^−6^	22
Alcoholism	0.117021277	2.104484053	0.026983909	22
ECM-receptor interaction	0.235955056	4.243362126	3.40346 × 10^−6^	21
Drug metabolism-cytochrome P450	0.164383562	2.956236628	0.026983909	12
**Pathway Upregulated in Type II**	**Rich Factor**	**Fold Enrichment**	**FDR**	**Count**
Complement and coagulation cascades	0.272727273	4.944785276	9.47878 × 10^−9^	24
Cytokine-cytokine receptor interaction	0.134228188	2.43367508	l.90717 × 10^−5^	40
TNF signaling pathway	0.151260504	2.742485951	0.007931472	18
Viral protein interaction with cytokine/cytokine receptor	0.16	2.900940695	0.007931472	16
Amoebiasis	0.145631068	2.640419322	0.029752513	15
IL-17 signaling pathway	0.147368421	2.671919061	0.033556746	14
Cytoskeleton in muscle cells TypeII	0.107758621	1.95375855	0.042656078	25

## Data Availability

Raw data for the five ccRCC PDX tissues have been deposited to the ProteomeXchange Consortium via the PRIDE partner repository (https://www.ebi.ac.uk/pride/ (accessed on 21 February 2025)) with the dataset identifier PXD061075. Supporting data files can be downloaded at https://stanfordmedicine.box.com/s/3mr7x89qw0pjffoifxj64efyb44kgt7u (accessed on 13 March 2025).

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
