# Peer review of "Identification of Molecular Subtypes of Clear-Cell Renal Cell Carcinoma in Patient-Derived Xenografts Using Multi-Omics"

_cancers, 2025, doi:10.3390/cancers17081361_

Round 1
Reviewer 1 Report
Comments and Suggestions for Authors
Authors has shown PDX resembles one of the human molecular subtypes closely in ccRCC, at both transcript and protein levels. It has been shown previously by many other groups in other cancer type as well. The results further emphasizes that PDX is the closet pre-clinical model we have got. However, I have following questions:
- If you have genomic data, will you see similar mutational profile? What was the mutational similarity? Were there any clonal differences? Will the subtype matching strategy hold in hyper-mutated tumor specimen?
- Line 141 “Significantly differentially expressed genes were identified using Python” does not make any sense to me. Could you list the package/library or provide the codes if custom script was used?
- Not sure, if only 3 sample is enough for subtype analysis.
- Was the mouse genome computationally removed from the PDX for the transcriptomic analysis ?
Best.
Author Response
1. If you have genomic data, will you see similar mutational profile? What was the mutational similarity? Were there any clonal differences? Will the subtype matching strategy hold in hyper-mutated tumor specimen?
Thank you for pointing this out. We agree with the reviewer that genetic mutations play a key role in the development, progression, prognosis, and treatment response of clear cell renal cell carcinoma (ccRCC) [1]. A recent study of 252 ccRCC patients by Ruan et al. identified g two molecular subtypes based on gene expression, DNA methylation, and gene mutation data [2]. Although no significant difference was observed for the mutation frequency of the most frequently mutated gene VHL between the two subtypes, other frequently mutated genes including PBRM1 and BAP1 were differentially mutated. In addition, no significant difference in TMB between the two subtypes was observed. We are in the process of generating genetic mutational profiles for our PDX models. It would be interesting to determine the mutational similarity of our PDX subtypes to ccRCC patient subtypes such as the negative correlation between the presence of PBRM1 and BAP1 mutants. We added this to the Discussion section on page 13 first paragraph line 395-405.
2. Line 141 “Significantly differentially expressed genes were identified using Python” does not make any sense to me. Could you list the package/library or provide the codes if custom script was used?
We revised the sentence to “Significantly differentially expressed genes were identified using the R packages DEGseq2” in the Methods section on page 4 line 149-150.
3. Not sure, if only 3 sample is enough for subtype analysis.
Thank you for pointing this out. We acknowledge that the size of our PDX models N=5 is small compared to the size of ccRCC patients for subtype analysis. Therefore, we focused on matching each PDX to one of the ccRCC patient subtypes rather than defining PDX subtypes. In general, two to three PDX models will be sufficient to investigate the mechanisms of progression and predict drug responses of a particular ccRCC subtype that the PDX models most resemble. Using the matching strategy, we hope our five PDX models will serve as realistic platforms in real-world situations to facilitate the clinical translation of findings to optimal management of ccRCC patients.
4. Was the mouse genome computationally removed from the PDX for the transcriptomic analysis?
Thank you for pointing this out. The mouse sequences were computationally removed from the PDX for the transcriptomic analysis using the R-package XenofilteR [3] available at https://github.com/NKI-GCF/XenofilteR. We added this information to the Methods section on page 4 line 145-147.
[1] Xie, D.; Li, G.; Zheng, Z.; Zhang, X.; Wang, S.; Jiang, B.; Li, X.; Wang, X.; Wu, G. The molecular code of kidney cancer: A path of discovery for gene mutation and precision therapy. Mol Aspects Med 2025, 101, 101335.
[2] Ruan, X.; Lai, C.; Li, L.; Wang, B.; Lu, X.; Zhang, D.; Fang, J.; Lai, M.; Yan, F. Integrative analysis of single-cell and bulk multi-omics data to reveal subtype-specific characteristics and therapeutic strategies in clear cell renal cell carcinoma patients. J Cancer 2024, 15, 6420-6433.
[3] Kluin, R.J.C.; Kemper, K.; Kuilman, T.; de Ruiter, J.R.; Iyer, V.; Forment, J.V.; Cornelissen-Steijger, P.; de Rink, I.; ter Brugge, P.; Song, J.-Y.; et al. XenofilteR: computational deconvolution of mouse and human reads in tumor xenograft sequence data. BMC Bioinformatics 2018, 19, 366.

Reviewer 2 Report
Comments and Suggestions for Authors
- The study fills an important niche by demonstrating that ccRCC PDXs not only retain transcriptomic and proteomic signatures of patient subtypes but also exhibit corresponding metabolic profiles.
- The use of bulk RNA-seq, LC-MS-based proteomics, and metabolomics provides a comprehensive assessment of tumor heterogeneity.
- The use of multiple independent human datasets (GSE3538, CPTAC, IM151/JR101) to anchor subtype definitions strengthens the generalizability of the results.
- The findings support the utility of subtype-matched PDXs in preclinical research and therapeutic testing, which is highly relevant for personalized medicine.
- While the clustering results are convincing, it would be helpful to further explain the biological implications of each subtype. For example, does subtype 1 (093/068/072) represent immune-cold or metabolically active tumors?
- Consider citing relevant literature on cancer initiation and progression, especially in the introduction, to frame the heterogeneity of ccRCC in the broader context of tumor evolution “Bhat AS, Ahmed M, Abbas K, Mustafa M, Alam M, Salem MAS, et al. Cancer Initiation and Progression: A Comprehensive Review of Carcinogenic Substances, Anti-Cancer Therapies, and Regulatory Frameworks. Asian J Res Biochem. 2024;14(4):111–125. doi:10.9734/ajrb/2024/v14i4300”.
- In Figure 4D and Table 2, the fold enrichment is described but the associated FDR or p-values for each pathway should be more clearly discussed in the text.
- The differences in lactate and leucine levels are intriguing. Could the authors speculate whether these differences may impact therapy response, especially with immunotherapy or metabolic inhibitors?
- "Muti-omics" should be corrected to "multi-omics" throughout.
- A few grammatical slips in the Introduction and Discussion should be cleaned up (e.g., "overlayed" → "overlapped").
- Some figure legends (e.g., Figure 6) would benefit from more explanation of statistical tests used and n-values.
- Tables S1–S4 are referenced but not provided. These should be made available to verify gene lists, metabolite values, and cluster assignments.
Recommendation
Minor Revision
The manuscript presents a technically sound and conceptually strong body of work. Minor revisions to expand interpretation, correct typographic issues, and clarify figure details would enhance its clarity and impact.
Author Response
1. The study fills an important niche by demonstrating that ccRCC PDXs not only retain transcriptomic and proteomic signatures of patient subtypes but also exhibit corresponding metabolic profiles.
We appreciate your encouraging comment, which is valuable for improving our work.
2. The use of bulk RNA-seq, LC-MS-based proteomics, and metabolomics provides a comprehensive assessment of tumor heterogeneity.
We appreciate your encouraging comment, which is valuable for improving our work.
3. The use of multiple independent human datasets (GSE3538, CPTAC, IM151/JR101) to anchor subtype definitions strengthens the generalizability of the results.
We appreciate your encouraging comment, which is valuable for improving our work.
4. The findings support the utility of subtype-matched PDXs in preclinical research and therapeutic testing, which is highly relevant for personalized medicine.
We appreciate your encouraging comment, which is valuable for improving our work.
5. While the clustering results are convincing, it would be helpful to further explain the biological implications of each subtype. For example, does subtype 1 (093/068/072) represent immune-cold or metabolically active tumors?
Thank you for pointing this out. We agree with you that the clustering results had biological implications of each subtype shown by GSEA analysis. Distinct pathways were enriched in differentially expressed genes between 093/068/072 vs. 054/047. For instance, multiple pathways involved in adhesion and extracellular matrix (ECM) modeling were upregulated in 093/068/072 including cell adhesion, focal adhesion and ECM-receptor interaction (Figure 4B), suggesting subtype 1 represents tumors that are highly active in tissue reorganization. In contrast, 054/047 showed high activities of immune response-related pathways including complement and coagulation cascades, cytokine-cytokine receptor interaction, TNF signaling, viral protein interaction with cytokine, and IL-17 signaling (Figure 4C), suggesting that subtype 2 encompasses immune-hot tumors. We added the text in red to the Results section on page 9, line 320-321, 324-325.
6. Consider citing relevant literature on cancer initiation and progression, especially in the introduction, to frame the heterogeneity of ccRCC in the broader context of tumor evolution “Bhat AS, Ahmed M, Abbas K, Mustafa M, Alam M, Salem MAS, et al. Cancer Initiation and Progression: A Comprehensive Review of Carcinogenic Substances, Anti-Cancer Therapies, and Regulatory Frameworks. Asian J Res Biochem. 2024;14(4):111–125. doi:10.9734/ajrb/2024/v14i4300”.
Thank you for pointing this out. We added this reference to the introduction on page 2 line 66.
7. In Figure 4D and Table 2, the fold enrichment is described but the associated FDR or p-values for each pathway should be more clearly discussed in the text.
Thank you for pointing this out. We discussed the FDR for top-ranking pathways in the result section. We added “ECM-receptor interaction and complement and coagulation cascades are the top pathways with the highest fold enrichment and the lowest FDRs, confirming the significance of these pathways in distinguishing the two PDX subtypes.” to page 9, line 326-328.
8. The differences in lactate and leucine levels are intriguing. Could the authors speculate whether these differences may impact therapy response, especially with immunotherapy or metabolic inhibitors?
Thank you for pointing this out. Lactate metabolism plays a critical role in ccRCC progression, where tumor cells preferentially rely on glycolysis rather than oxidative phosphorylation for energy production, resulting in lactate accumulation even in the presence of oxygen [4]. Lactate also serves as a signaling molecule that activates oncogenic pathways and promotes angiogenesis by stabilizing HIF, further driving tumor progression [5]. Lactate can inhibit the function of immune cells, such as T cells and natural killer cells, promoting immune evasion and creating an immunosuppressive environment that supports tumor growth [6]. High lactate level in subtype 2 tumors indicates the potential to target lactate metabolism as a therapeutic strategy to disrupt the metabolic and signaling pathways that sustain their growth and progression as well as potentiate anti-angiogenic and immunotherapy. Interestingly, subtype 2 also showed a high level of branched-chain amino acids (BCAAs) including leucine, isoleucine, and valine, that have been shown to have double-edge effects in cancer progression [7,8] On the one hand, BCAAs are indispensable for maintaining immune system function, and dietary supplementation with BCAAs can enhance the activity of immune cells [9,10]. For example, leucine is essential for the upregulation of mTORC1 during T cell activation [11]. On the other hand, BCAAs can serve as an energy source and an activator of oncogenic signaling pathways, potentially fueling cancer development by promoting cell proliferation, migration, and invasion [12]. Therefore, the utility of BCAAs including Leu, either as a biomarker or as a potential therapeutic target, in ccRCC will depend on further research clarifying their effects. We added this to the Discussion on page 13, line 406-426.
9. "Muti-omics" should be corrected to "multi-omics" throughout.
Thank you for pointing this out. We corrected it on page 1 line 3 and page 2 line 81.
10. A few grammatical slips in the Introduction and Discussion should be cleaned up (e.g., "overlayed" → "overlapped").
Thank you for pointing this out. We corrected it on page 7 line 281.
11. Some figure legends (e.g., Figure 6) would benefit from more explanation of statistical tests used and n-values.
Thank you for pointing this out. We added N-values and the statistical test used to the legend of Figure 6 on page 12 line 365-367.
12. Tables S1–S4 are referenced but not provided. These should be made available to verify gene lists, metabolite values, and cluster assignments.
Thank you for pointing this out. We stated in the supplementary material section that Tables S1-S4 are available at the following link: https://stanfordmedicine.box.com/s/3mr7x89qw0pjffoifxj64efyb44kgt7u.
We also provided the link in the data availability statement section.

Reviewer 3 Report
Comments and Suggestions for Authors
Qui and colleagues have classified 5 ccRCC PDX models based on their transcriptomic and proteomic profiles, and compared them to subtypes identified in several cohorts of human ccRCC. They found that each PDX was similar in molecular profile to at least one of the different ccRCC subtypes identified in the patient cohorts that they interrogated. They propose that a PDX subtype that mirrors the human ccRCC subtype under investigation should be used to investigate molecular mechanisms of cancer progression and for optimizing therapies for ccRCC patients. The paper is well written with robust PDX-human ccRCC comparative omics data. The conclusions are well supported by the data presented.
Please address the following points:
- The authors did not compare the PDX omics profiles to that of the TCGA ccRCC profiles. This is a large publicly available data set and a rich resource. Was there a reason this cohort was not used in these comparisons?
- To underscore the point of similarity among the 093, 068 and 072 PDX models and, on the other hand, the more close association of 047 and 054 PDXs, the authors might set up an efficacy study with a HIF2 or VEGF inhibitor and compare the tumor responses in the two PDX subtype groups. One might predict similar responses among the PDX models within each subgroup and it would be interesting to test that out.
- It would be important for the authors to discuss how this “matching” strategy might be used in real world situations to facilitate the clinical translation of findings to optimal management of ccRCC patients.
- Line 265-should this be 054 and 047 not 072 and 047?
- Some references are not complete.
Author Response
1. The authors did not compare the PDX omics profiles to that of the TCGA ccRCC profiles. This is a large publicly available data set and a rich resource. Was there a reason this cohort was not used in these comparisons?
Thank you for pointing this out. We agree with the reviewer that TCGA ccRCC dataset is a rich resource. However, compared to the datasets we selected for this study, it lacks critical information. For example, it does not have proteomics data compared to CPTAC PDC000200. It also does not have treatment response data compared to IM151 and JR101 trials. We used GSE3538 because it is an in-house dataset and the number of patients in the cohort is big enough for our study purpose.
2. To underscore the point of similarity among the 093, 068 and 072 PDX models and, on the other hand, the more close association of 047 and 054 PDXs, the authors might set up an efficacy study with a HIF2 or VEGF inhibitor and compare the tumor responses in the two PDX subtype groups. One might predict similar responses among the PDX models within each subgroup and it would be interesting to test that out.
Thank you for pointing this out. We agree with the reviewer that an efficacy study with an HIF2 or VEGF inhibitor would be appropriate to determine whether the two PDX subtypes respond differently and whether PDX lines within the same subtype would respond similarly. It would be interesting to test whether the two subtypes respond differently to other standard-of-care treatments such as immune checkpoint inhibitors. Our results provide a strong rationale for such studies in the future which may help the research community to select the right models for investigating mechanisms of treatment response and resistance in ccRCC. We added this to the Discussion on page 14 line 457-464.
3. It would be important for the authors to discuss how this “matching” strategy might be used in real world situations to facilitate the clinical translation of findings to optimal management of ccRCC patients.
Thank you for pointing this out. The matching strategy will allow more accurate preclinical testing and personalized treatment strategies in ccRCC patient care. Different PDX subtypes are ideal models for identifying effective treatments and biomarkers for treatment responses for each patient subtype they resemble. In real world, when a patient is assigned to a molecular subtype, he/she can be treated with the most effective treatment(s) identified in the same PDX subtype. In addition, PDX subtypes can be used to test novel drugs and delivery methods to identify effective new treatments for corresponding ccRCC patient subtypes. When multiple PDX lines in the same subtype are used, it mimics a clinical trial conducted in the corresponding ccRCC patient subtype. These applications will greatly facilitate the clinical translation of findings and improve ccaRCC patient outcomes. We added this to the discussion section on page 14 line 448-456.
4. Line 265-should this be 054 and 047 not 072 and 047?
Thank you for pointing this out. We corrected it.
5. Some references are not complete.
Thank you for pointing this out. We checked all references to ensure they are complete.

Reviewer 4 Report
Comments and Suggestions for Authors
Overall
This paper demonstrated that each of the authors developed PDX models with various subtypes of RCC. The concept of research is valuable and the result is very interesting. But, I have several concerns and questions.
Major
- Abstract Objective was not described clearly.
- Table 1 What is the selection criteria of PDXs?
- Page 5 ccRCC PDX metabolic data, What is the data of unit? Are they calculated data?
- Page 7 Figure 2 and Page 8 Figure 3; Why are many points plotted?
Author Response
1. Abstract Objective was not described clearly.
Thank you for pointing this out. We revised the abstract and added “Our objective is to compare the transcriptional and proteomic profiles of our PDX models to those of ccRCC patients and identify both similarities and distinctions between molecular profiles of PDX subtypes and corresponding ccRCC patient subtypes, so that proper PDX subtypes can be used when investigating the corresponding ccRCC patient subtypes” on page 1 line 33-37.
2. Table 1 What is the selection criteria of PDXs?
Thank you for pointing this out. We selected PDXs that had multi-omics data including transcriptomic, proteomic, and metabolic profiles for this study. We added this to the Methods on page 3 line 138-139.
3. Page 5 ccRCC PDX metabolic data, What is the data of unit? Are they calculated data?
Thank you for pointing this out. The unit is nmol/mg. They are absolute concentrations. We added this to the method section on page 5 line 208.
4. Page 7 Figure 2 and Page 8 Figure 3; Why are many points plotted?
Thank you for pointing this out. These are PCA analyses comparing two groups of samples. Specifically, Figure 2C includes 47 PDX samples and 177 ccRCC patient samples (GSE3538). Figure 3B compares the 177 ccRCC patients with another cohort of 110 ccRCC patients (CPTAC PDC000200) and Figure 3C has the 47 PDX samples and the 110 ccRCC patient samples.

Round 2
Reviewer 3 Report
Comments and Suggestions for Authors
The authors have addressed the reviewer's comments in a satisfactory manner. This improved manuscript does not require any further editing.
Reviewer 4 Report
Comments and Suggestions for Authors
I have confirmed all responses and revised article. The paper was revised by points by points. ANd the paper was improved in the view of scientific soundness.
I have no more comments for improvement.